# UC-DiffOSI: Universal Controllers with Differentiable Physics for Online System Identification

## Abstract

Creating robots that can handle changing or unknown environments is a critical step towards real-world robot applications. Existing methods tackle this problem by training controllers robust to large ranges of environment parameters (Domain Randomization), or by combining "Universal" Controllers (UC) conditioned on environment parameters with learned identification modules that (implicitly or explicitly) identify the environment parameters from sensory inputs (Domain Adaptation). However, these methods can lead to over-conservative behaviors or poor generalization outside the training distribution. In this work, we present a domain adaptation approach that improves generalization of the identification module by leveraging prior knowledge in physics. Our proposed algorithm, UC-DiffOSI, combines a UC trained on a wide range of environments with an Online System Identification module based on a differentiable physics engine (DiffOSI). We evaluate UC-DiffOSI on articulated rigid body control tasks, including a wiping task that requires contact-rich environment interaction. Compared to previous works, UC-DiffOSI outperforms domain randomization baselines and is more robust than domain adaptation methods that rely on learned identification models. In addition, we perform two studies showing that UC-DiffOSI operates well in environments with changing or unknown dynamics. These studies test sudden changes in the robot's mass and inertia, and they evaluate in an environment (PyBullet) whose dynamics differs from training (NimblePhysics).

## 1 Introduction

In order for robots to shine in real-world applications, they need to handle ever-changing and unpredictable situations in real environments. For instance, a robot waiter should be able to serve a new type of dish without spilling food, and an autonomous vehicle should take a person safely to an unvisited destination. Creating artificial agents that can operate in changing and unknown environments is a longstanding problem in the robotics community.

The collective wisdom of the robotic research community in recent years indicates that enabling learning agents to work in changing and unknown environments is not about making one big breakthrough, but rather making many small but informed decisions. One general approach advances control policies such that they can operate more robustly (Tan et al., 2018) or more adaptively (Cully et al., 2015) in testing environments. However, these methods usually exhibit sub-optimal task performance or require additional fine-tuning in the target environment. Alternative approaches advance simulation techniques to bring the training environment closer to the testing one prior to learning a control policy, such as training a dynamics model (Jiang et al., 2021) or identifying simulation parameters (Tan et al., 2018) from data. These methods are often used in offline settings as learning or identifying an accurate simulation model can be time consuming. Thus, they cannot handle changing environments naturally. Work such as Yu et al. (2017) have also investigated methods that learn models to perform online system identification. However, a learned model often does not generalize well to unknown environments or those not seen during training.

Recent developments in differentiable physics simulation potentially offer a more effective way to address these challenges by advancing both control and simulation techniques. By utilizing fast

computation of analytical gradients, one can devise more computationally and sample efficient optimal control and system identification algorithms. Recent differentiable physics simulators, such as NimblePhysics (Werling et al., 2021), provide fast computation of analytical gradients in the face of constraint satisfaction and non-differentiable contact handling. These enable generic gradient-based optimizers to solve contact-rich optimal control problems. While promising, differentiable physics simulation does not solve the fundamental problem of multiple local minima due to ill-conditioned cost functions, often exacerbated by long-horizon and highly nonlinear differential equations.

This paper introduces a new approach for creating resilient and adaptive agents by combining differentiable physics simulation for *online* system identification and reinforcement learning for *offline* policy training. Online system identification can be formulated as a short-horizon, local optimization problem, but must be solved fast. This plays to the strength of differentiable physics simulation which provides analytical gradients efficiently, while avoiding the pitfalls of poor cost function landscapes. On the other hand, for challenging control problems with long-horizon cost functions, we resort to a reinforcement learning approach leveraging samples ("rollouts") generated offline at scale to train a control policy. We explore many possible situations the agent might encounter when operating in the testing environment by varying the simulation parameters during training and learning a *Universal Controller* (UC) conditioned on the simulation parameters. At test time, we use differentiable physics simulation to continuously optimize the simulation parameters based on the most recent history of observations (DiffOSI). The optimal simulation parameters will "modulate" the universal policy to output the optimal action for the currently identified environment.

We evaluate our approach on two robotic control tasks, a cartpole balancing problem and a robot arm table wiping task involving rich contact phenomena. We show that our proposed approach combining a Universal Controller and a Differentiable physics-based Online System Identification module (UC-DiffOSI) can outperform pure learning-based or traditional system identification methods. Finally, we demonstrate that our approach can be applied to environments with changing dynamics or un-modeled effects.

## 2    RELATED WORK

**Deep Reinforcement Learning and Domain Randomization.**  Deep reinforcement learning has been proven to be effective in learning complex motor skills for simulated robots, such as running (Yu et al., 2018), parkour (Heess et al., 2017), and dressing (Clegg et al., 2018). However, these controllers often perform poorly on real robot hardware due to the discrepancies between the simulated and real environment, also known as the sim-to-real gap (Neunert et al., 2017). Domain randomization of the simulation physics parameters has been extensively explored to help the simulation-trained controller transfer to a different target environment, where a robust control policy is trained to perform well for a wide variety of simulated environments (Peng et al., 2017; Tan et al., 2018; Hwangbo et al., 2019; Exarchos et al., 2020; OpenAI et al., 2019). However, policies trained from domain randomization often exhibit over-conservative behaviors, leading to sub-optimal performance (Tan et al., 2018). Different from these methods, we develop a domain adaptation approach by training adaptive controllers that can adjust behavior for different environments using an estimation of the environment parameters. This enables our controller to achieve better performance than a domain randomization controller.

**Domain Adaptation.**  To achieve better task performance in novel situations, researchers have developed adaptive controllers that can adjust behavior for different environments. Szita et al. (2003) showed that Q-learning, using event-learning, can find near-optimal policies in varying environments. Heess et al. (2015) demonstrated that control policies modified to use recurrent networks are also capable of dealing with unknown kinematic parameters such as link lengths. Xu et al. (2020) presented a deep reinforcement learning method that encodes the dynamic context online to achieve a stable non-planar pushing task controller. Yu et al. (2017) proposed a system using a Universal Policy and Online System Identification (OSI) function to explicitly incorporate model parameters to adapt to varying environments. These methods usually identify the environment parameters (explicitly or implicitly) and then adjust the controllers to adapt to the new environment.

**Differentiable Simulation.**  In recent years, researchers have built more efficient and feature-complete differentiable physics engines. These engines support 3D rigid body and contact constraints between spheres and planes (Degrave et al., 2016), analytic differentiation of a linear com-

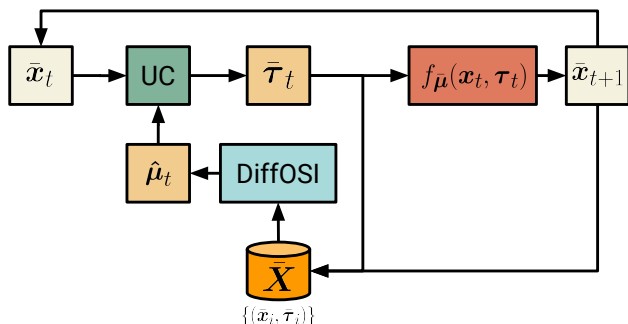

Figure 1: Overview of our method, UC-DiffOSI. The Universal Controller (UC) takes as input the current robot states $x_t$ and the dynamics parameters $\hat{\mu}_t$ identified by the differentiable physics engine (DiffOSI), to generate optimal control actions $\tau_t$.

plementarity problem (de Avila Belbute-Peres et al., 2018), modeling soft bodies via a differentiable real-time differentiable Material Point Method (Hu et al., 2018), support differentiable cloth simulation (Liang et al., 2019), optimize for large numbers of objects and contact interactions (Qiao et al., 2020), and support articulated rigid bodies with contact (Werling et al., 2021). Prior work, such as Toussaint et al. (2018) and Heiden et al. (2019), also showed that differentiable physics can be integrated for end-to-end controller learning, in addition to parameter learning. Jatavallabhula et al. (2021) further integrated differentiable rendering to remove dependency on 3D vision in an end-to-end learning pipeline.

## 3  METHODS

Our goal is to design a system that can handle changing or unknown dynamics in the environment. The true dynamics in the target environment can be described by $x_{t+1} = f_{\mu}(x_t, \tau_t)$, where $x = (q, \dot{q})$ denotes the robot's sensed states and their time derivatives, and $\tau$ denotes the control actions. $f_{\mu}$ evolves the system from timestep $t$ to $t + 1$ with dynamics parameters $\mu$. We aim to predict optimal controls $\tau_t^*$ which maximize task performance. The controls are predicted in the first part of our system, the universal controller (UC): $(x, \mu) \mapsto \tau$. The second part is a differentiable physics engine that performs online system identification (DiffOSI): $\{(x_i, x_{i+1}, \tau_i)\} \mapsto \mu$. The overview of the system is shown in Figure 1. Together, they form a robust controller capable of handling unknown or changing environment dynamics.

### 3.1  LEARNING A UNIVERSAL CONTROLLER

Universal Controller (UC) augments a regular robotic controller by conditioning it on parameters of the environment $\mu$, such as friction coefficient or robot payload. This information is crucial for the controller to select appropriate actions for different environments, yet are non-trivial to infer directly from sensory input. By providing this additional information to the UC, we expect it to outperform a regular policy given the true environment parameters.

A successful UC should perform near-optimally for a wide range of $\mu$'s. Given that the best way to obtain a control policy can be different across tasks, the training of UC largely depends on the task to be performed. In this work, we tailor the training of UC to two control tasks of interest: cart pole balancing and table wiping. For the cart-pole balancing problem, we want to obtain a controller that directly sends torque commands to the robot at high-frequency. Thus, we directly apply a reinforcement learning approach to obtain a Universal Control Policy as done in Yu et al. (2017). On the other hand, for the table wiping problem, we adopt a hierarchical control structure where the learned UC needs to modulate the parameters of a low-level admittance controller per wiping motion. A black-box optimization technique is more suitable for this low-frequency problem. More details on how we train our UPs can be found in Section 4.

Note that we train our UC with a set of training environments $g_{\hat{\mu}}$, which approximates the target environment $f_{\bar{\mu}}$. $\bar{\mu}$ and $\hat{\mu}$ need not represent the same set of parameters, and neither do $f_{\bar{\mu}}$ and

$g_{\hat{\mu}}$ need to represent the same model, as the exact governing equations of $f_{\bar{\mu}}$ is usually unknown. We aim to use a $g_{\hat{\mu}}$ diverse enough to train a robust UC and expressive enough to approximate all possible trajectories evolved with $f_{\bar{\mu}}$.

## 3.2 DIFFERENTIABLE PHYSICS FOR ONLINE SYSTEM IDENTIFICATION

We use differentiable physics to identify some unknown physics parameters $\hat{\mu} \in \mathbb{R}^g$ that parameterize the dynamics of the system. The numerical modeling, $\boldsymbol{x}_{t+1} = g_{\hat{\mu}}(\boldsymbol{x}_t, \boldsymbol{\tau}_t)$, in the differentiable physics engine should be the same as in UC training approximating the target environment dynamics, $\boldsymbol{x}_{t+1} = f_{\bar{\mu}}(\boldsymbol{x}_t, \boldsymbol{\tau}_t)$. This way, the nature of the UC's inputs stays consistent.

To perform system identification, DiffOSI requires first collecting a small number of samples $\bar{X} = \{\bar{\boldsymbol{x}}_t, \bar{\boldsymbol{\tau}}_t\}$ from target environment $f_{\bar{\mu}}$. DiffOSI uses these samples to optimize for a $\hat{\mu}$ that minimizes the differences between the resulted trajectory $\{\hat{\boldsymbol{x}}_t, \hat{\boldsymbol{\tau}}_t\}$ and the target state-action history $\{\bar{\boldsymbol{x}}_t, \bar{\boldsymbol{\tau}}_t\}$. DiffOSI requires a minimum of two samples, but if the problem is nondeterministically underconstrained (e.g., in the presence of contact), more samples (e.g., 30-50) may be required to exercise all dynamics of the system.

At the beginning (first iteration $k = 0$), we initialize $\hat{\mu} = \mu_0$ (e.g., mean of expected distribution). For each iteration of DiffOSI optimization $k$, we execute the UC for a certain number of steps $T_k \le |\bar{X}_k|$ using actions predicted with $UC(\bar{\boldsymbol{x}}_k, \hat{\mu}_k)$. With the collected samples $\bar{X}_k$, we use DiffOSI to optimize for the $\hat{\mu}_k$ that minimizes the following objective function:

$$\mathcal{L}(\bar{X}_k) = \sum_{t \in \bar{X}_k} \phi(\hat{\boldsymbol{q}}_{t+1}, \bar{\boldsymbol{q}}_{t+1}) + \phi(\dot{\hat{\boldsymbol{q}}}_{t+1}, \dot{\bar{\boldsymbol{q}}}_{t+1}), \tag{1}$$

$$\text{where } (\hat{\boldsymbol{q}}_{t+1}, \dot{\hat{\boldsymbol{q}}}_{t+1}) = g_{\hat{\mu}_k}(\bar{\boldsymbol{q}}_t, \dot{\bar{\boldsymbol{q}}}_t, \bar{\boldsymbol{\tau}}_t) \tag{2}$$

and $\phi$ is any differentiable distance function.

A differentiable physics engine enables the computation of gradients of $\mathcal{L}$ with respect to the unknown parameters $\mu$:

$$\frac{\partial \mathcal{L}(\bar{X}_k)}{\partial \hat{\mu}_k} \tag{3}$$

Our system is agnostic to the choice of the differentiable physics engine. We use the Nimble differentiable physics engine (based on DART) by Werling et al. (2021), which has the advantage of being able to handle articulated rigid bodies and differentiate through contact.

## 4 EXPERIMENTS

### 4.1 TASK EVALUATION OVERVIEW

We compare our proposed algorithm (UC-DiffOSI) to six baseline methods:

1. **Domain Randomization (DR):** Optimize a controller in an environment where dynamics parameters are randomized.
2. **UC-Random:** UC given random parameters as input.
3. **UC-Average:** UC given the middle parameter of the training range as input.
4. **UC-MLP (Yu et al.):** UC given parameters predicted with an MLP.
5. **UC-CMA-ES:** UC given parameters optimized using CMA-ES (Hansen, 2016).
6. **UC-Oracle:** UC given ground truth parameters as input.

to address the following questions:

- Does UC-DiffOSI outperform Domain Randomization (DR)?
- Is DiffOSI more robust than MLP, in generalization to new environments?
- Is DiffOSI more efficient than CMA-ES?

Table 1: Results on the Cartpole task. We report errors on the parameter estimation of $\mu$ (lower is better), as well as the overall task performance (higher is better). Mean and standard deviation are reported. Results are averaged over 3 models and 10 episodes. Task performance is defined as the number of simulation steps where the state of the cartpole is within certain thresholds.

| Approach | Mean Abs. Error ($\mu$) | Mean Abs. Rel. Error ($\mu$) | Task Performance |
|---|---|---|---|
| DR | N/A | N/A | $277.77 \pm 169.63$ |
| UC-Random | $0.45 \pm 0.11$ | $1.42 \pm 1.50$ | $169.63 \pm 127.92$ |
| UC-Average | $0.39 \pm 0.15$ | $0.50 \pm 0.00$ | $215.83 \pm 126.89$ |
| UC-MLP-Narrow | $0.32 \pm 0.12$ | $0.60 \pm 0.13$ | $242.77 \pm 157.96$ |
| UC-MLP | $0.09 \pm 0.03$ | $0.38 \pm 0.43$ | $309.23 \pm 72.55$ |
| UC-DiffOSI (Ours) | $0.09 \pm 0.07$ | $0.13 \pm 0.05$ | $390.77 \pm 95.67$ |
| UC-Oracle | $0.00 \pm 0.00$ | $0.00 \pm 0.00$ | $451.33 \pm 48.59$ |

We compare our proposed algorithm with baseline methods in both task performance (total toward over one episode) as well as prediction accuracy for the environment parameters (mean absolute error and mean absolute relative error).

## 4.2 CARTPOLE

**Task Description.** We first evaluate our algorithm on the cartpole task from OpenAI Gym (Brockman et al., 2016), where the goal is to balance a pole on a cart-on-track. The state space of the cartpole task consists of the position of the cart and pole $q = (x, \theta)$, as well as their velocities $\dot{q} = (\dot{x}, \dot{\theta})$. To create different environment variations, we offset the center of mass of the pole from its geometric center by a random displacement $(\mu, 0.2\mu)$ in 2D, with $\mu$ uniformly sampled within the range $[-0.6m, 0.6m]$. This is to mimic attachments of different objects on the pole We randomly initialize the $q$ and $\dot{q}$ of the cart and the pole, with $x \in [-0.05, 0.05], \dot{x} \in [-0.05, 0.05], \theta \in [-0.05, 0.05], \dot{\theta} \in [-0.05, 0.05]$. The controller can apply an impulse of $[-500N, 500N]\Delta t$ to the cart at 50 Hz. We define the task performance metric as the number of simulation steps where $|\theta| \leq \pi/2$ (where $\theta = 0$ means the pole is upright) and $|x| \leq 2.5$ are satisfied.

**Controller Details.** As a classic control problem, there a numerous ways to design a controller for the cartpole task. In this work, we leverage a reinforcement learning-based approach. Specifically, we use Proximal Policy Optimization (PPO) (Schulman et al., 2017) to train a control policy (MLP with 2 layers, 64 hidden units each) that takes the cartpole state $s = (q, \dot{q})$ as input, and predicts the appropriate control force applied to the cart.

For the domain randomization (DR) baseline, we train the control policy with $\hat{\mu}$ randomly selected at the beginning of each training episode. We then augment the policy input with $\hat{\mu}$ to obtain the universal controller (UC). By providing this critical information about the environment, the policy can better decide the optimal action to take, which is demonstrated in our evaluation results.

**System Identification.** For all methods, we use the observations containing $(q_t, \dot{q}_t, \tau_t, q_{t+1}, \dot{q}_{t+1})$. For the UC-MLP baseline, we train an MLP (3 hidden layers with 256, 128, and 64 hidden units respectively) to predict $\hat{\mu}$ from the states and actions in the history. We use a dataset collected by executing $\hat{\tau} = UC(x, \hat{\mu})$ in the cartpole simulation environment $g_{\hat{\mu}}$ for uniformly sampled $\mu$. Both UC-CMA-ES and UC-DiffOSI optimize $\hat{\mu}$ to minimize the MSE between the result and target trajectories.

**Results.** Results for the cartpole task can be found in Table 1. Our approach achieves better results than the baseline methods in terms of task performance and is close to the upper bound (UC-Oracle). This suggests our algorithm can successfully infer the environment parameters from the input history of past observations and actions. This is further supported by the low prediction error of our algorithm. We observe that UC-MLP can achieve low mean absolute error in predicted $\hat{\mu}$, yet performs worse in relative prediction error. This is possibly due to the model learning to focus on regions where the true $\hat{\mu}$ has higher magnitude, which also leads to inferior task performance. This can possibly be mitigated by tuning the loss function or training data further, yet these are task spe-

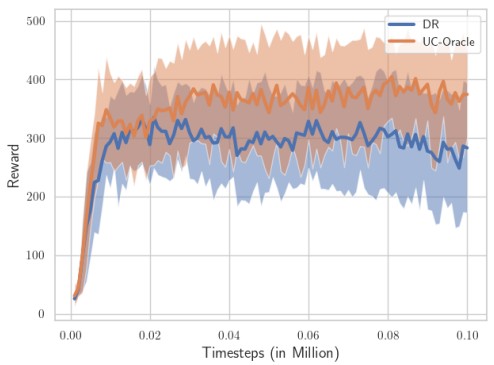 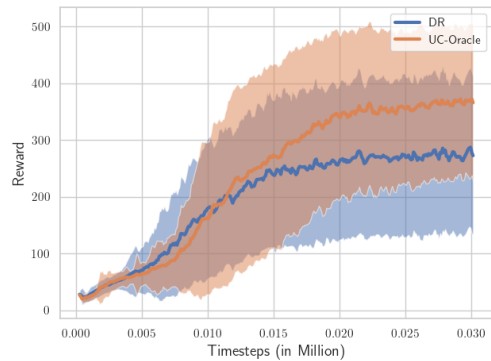

(a) Evaluation curves in the Nimble physics engine (averaged across 20 runs).

(b) Sim-to-Sim Transfer results in the Bullet physics engine (averaged across 20 runs).

Figure 2: Evaluation curves for the DR and UC-Oracle models on the Cartpole task.

cific, non-generalizable processes. In addition, we also evaluate MLP-Narrow, which trains an MLP model to predict the environment parameter, but with $\hat{\mu}$ sampled from a narrower range ($[-0.2, 0.2]$). As shown in Table 1, MLP-Narrow does not generalize to parameters outside the training range.

### 4.3 HALF CHEETAH

In this section, we evaluate our method on a 7-link, 6-DoF 2D half cheetah task (Figure 3a).

**Task Description.** We adapt the half cheetah environment from OpenAI Gym (Brockman et al., 2016), and replace the simulator with the NimblePhysics simulator. The mass of the half cheetah's torso is unknown, and uniformly sampled within the range $[1, 50]$ kg (where the original torso mass is 4.9 kg).

**Controller Details.** Similar to cartpole, we use the PPO algorithm to train a control policy (MLP with 2 layers, 256 hidden units each). We train 6 different seeds for each approach, DR and UC, where UC is trained with ground truth $mu true$, corresponding to the UC-Oracle results.

**System Identification.** Both the MLP and DiffOSI methods for system identification use the same inputs as those used in the cartpole task in Section 4.2. The MLP (2 hidden layers with 512 and 128 hidden units respectively) is trained on a dataset of 600K examples generated from rolling out trajectories from the trained UC-Oracle models over uniformly sampled $\mu$. The DiffOSI module uses NimblePhysics as its simulator, and optimizes $\hat{\mu}$ (initialized to the mean of the distribution) to minimize the L2 norm between the predicted and target next state.

**Results.** In Figure 3b, we show that UC-Oracle is able to outperform pure domain randomization. This shows that observation of physical parameters is beneficial for this task. Evaluations of various approaches using the trained DR and UC policies are shown in Table 2. We show that UC-DiffOSI is able to perform comparably to the upper bound, UC-Oracle, due to its low parameter estimation errors. UC-MLP has, on average, higher parameter estimation errors than UC-DiffOSI, but lower parameter estimation errors than random or average baselines (Figure 3c). However, we found that for some trajectory rollouts, UC-MLP errors tend to magnify over time (Figure 3d is one such example). This shows the weakness of UC-MLP – the UC depends on a perfect MLP for system identification, and any slight mistake the MLP makes can quickly cause errors to compound over time. These examples can lead to an even larger scale of failure at the task performance level (relative to the scale of $\mu$ estimation errors). Thus the overall task performance of UC-MLP is worse than UC-Random and UC-Average.

We also analyze the runtime of UC compared to an MPC baseline. To achieve reasonable task performance, MPC baseline needs to have a planning horizon of more than 200 timesteps and replan every 40 timesteps. Each trajectory optimization in MPC takes 100-300 milliseconds to solve while evaluating UC for 40 timesteps only takes roughly 40 milliseconds. As such, MPC is not a suitable method for a setting like ours where realtime performance is important.

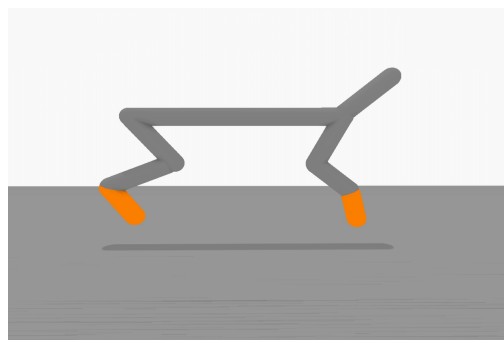

(a) The 7-link, 6-DoF 2D half cheetah task, simulated in NimblePhysics.

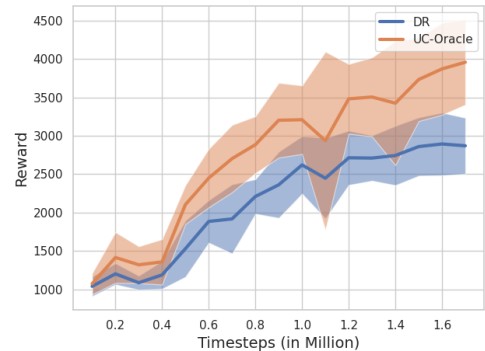

(b) Evaluation curves comparing DR and UC-Oracle (10 seeds).

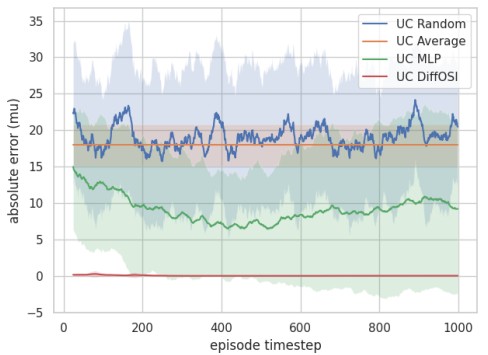

(c) Average absolute $\mu$ estimation error across episode timesteps (3 seeds).

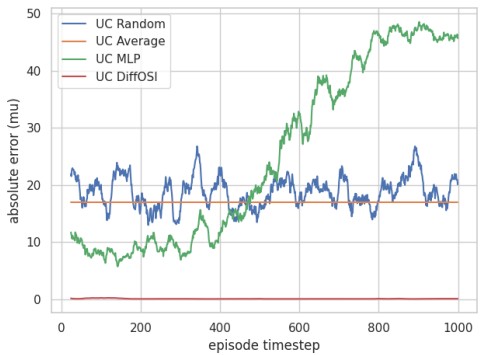

(d) Absolute $\mu$ estimation error on a single example across episode timesteps.

Figure 3: Reward and $\mu$ estimation results for the half cheetah task. Results are averaged over 6 models for each approach.

| Approach | Mean Abs. Error ($\mu$) | Mean Abs. Rel. Error ($\mu$) | Task Performance |
|---|---|---|---|
| DR | N/A | N/A | $2546.51 \pm 1593.81$ |
| UC-Random | $19.11 \pm 12.97$ | $2.27 \pm 3.42$ | $3567.45 \pm 588.88$ |
| UC-Average | $17.97 \pm 2.85$ | $2.24 \pm 2.63$ | $3596.58 \pm 802.73$ |
| UC-MLP | $9.25 \pm 10.20$ | $1.27 \pm 2.25$ | $3102.02 \pm 2202.76$ |
| UC-DiffOSI | $0.06 \pm 0.14$ | $0.01 \pm 0.02$ | $3930.97 \pm 801.29$ |
| UC-Oracle | $0.0 \pm 0.0$ | $0.0 \pm 0.0$ | $3936.30 \pm 609.14$ |

Table 2: Results on the Half Cheetah task (trained on 6 seeds each for DR and UC, each model evaluated on 3 random seeds).

## 4.4 TABLE WIPING

**Task Description.** In this task, we wipe a tabletop using a wiping tool attached to a robot arm's end effector, as shown in supplementary Figure 5a. The wiping tool joint has unknown stiffness and damping coefficients $\mu = \{(k_p, k_d)\}$. To perform a wipe, we track a scripted wiping trajectory $\{q_t^{traj}\}$ with an admittance controlled robot arm. The goal is to maintain a target contact force on the contact normal direction $f_{z,goal}$ during the whole wiping process, i.e., to minimize the MSE between $f_{z,sensed}$ and $f_{z,goal}$ for all observed states when the wiping tool is in contact with the tabletop. The system overview can be found in supplementary Figure 5b with details as follows.

Table 3: Results on the wiping task. We report errors on the parameter estimation of $\mu$ (lower is better), as well as the overall task performance (lower is better). Mean and standard deviation are reported. Results are averaged over 10 runs.

| Approach | Mean Abs. Error ($\mu$) | Mean Abs. Rel. Error ($\mu$) | Task Performance |
|---|---|---|---|
| DR | N/A | N/A | $1.69 \pm 1.58$ |
| UC-Random | $39.038 \pm 22.157$ | $3.607 \pm 7.874$ | $1.07 \pm 0.78$ |
| UC-Average | $22.296 \pm 12.333$ | $2.120 \pm 4.129$ | $0.92 \pm 0.73$ |
| UC-MLP | $36.875 \pm 20.324$ | $3.508 \pm 6.569$ | $0.93 \pm 0.75$ |
| UC-CMA-ES | $0.418 \pm 0.502$ | $0.029 \pm 0.037$ | $0.50 \pm 0.32$ |
| UC-DiffOSI | $0.001 \pm 0.001$ | $4.3\text{e-}05 \pm 4.0\text{e-}05$ | $0.51 \pm 0.32$ |
| UC-Oracle | $0.000 \pm 0.000$ | $0.000 \pm 0.000$ | $0.51 \pm 0.32$ |

**Controller Details.** The task controller is defined by the combination of the admittance control law and the admittance control parameters $\boldsymbol{\theta} = (\boldsymbol{k_\tau}, \boldsymbol{k_\xi})$.

For the DR baseline, we optimize for the optimal admittance control parameters $\boldsymbol{\theta}^*$ in the simulation environment $g_{\hat{\boldsymbol{\mu}}}$ .

The goal of the UC is to apply admittance control with the optimal control parameters mapped from the identified $\hat{\boldsymbol{\mu}}$. We collect a dataset (85 examples) to train the DNN-parameterized UC (2 hidden layers, 512 units each with ReLU activation functions and a linear FC layer). The dataset contains paired examples of $\hat{\boldsymbol{\mu}}_{true}$ and corresponding optimal control parameters $\boldsymbol{\theta}^* = (\boldsymbol{k_\tau^*}, \boldsymbol{k_\xi^*})$.

**System Identification.** Our goal with system identification for the wiping task is to predict dynamics parameters $\hat{\boldsymbol{\mu}}$ to condition the UC. For all approaches with UC, we first collect a trajectory in the environment (parameterized by some unknown $\bar{\boldsymbol{\mu}}$) using canonical admittance control parameters $\tilde{\boldsymbol{\theta}}$. From the collected trajectory, we uniformly sample 50 segments from the history to perform system identification, where each segment contains $(\boldsymbol{q}_t, \boldsymbol{q}_{t+1}, \boldsymbol{q}_t^{traj})$.

Both UC-CMA-ES and UC-DiffOSI optimize for the $\hat{\boldsymbol{\mu}}$ that minimize the MSE between the resulted and target sampled segments from the collected trajectory.

For UC-MLP, we feed the same 50 segments to the MLP and predict $\hat{\boldsymbol{\mu}}$. The MLP (same architecture as UC) is trained on a dataset mapping trajectory segments to $\hat{\boldsymbol{\mu}}_{true}$. The $\hat{\boldsymbol{\mu}}_{true}$ is uniformly sampled, and the trajectories are generated by executing the UC with $\hat{\boldsymbol{\mu}}_{true} \mapsto \hat{\boldsymbol{\theta}}$.

**Results.** We evaluate all baselines on the wiping task and report our results in Table 3. For reference, we also include UC using random or average $\hat{\boldsymbol{\mu}}$ as baselines. UC with average $\hat{\boldsymbol{\mu}}$ can be seen as an offline system identification method. For simulation, we use a timestep of 1e-3s. For all our experiments, we set $F_{z,goal} = 3$N and uniformly sample $\bar{\boldsymbol{\mu}} \in [1, 100]$. DR performs worse than all UC-based approaches, which indicates that UCs provide useful physics-awareness for the wiping task. UC-MLP performs slightly better. Both UC-CMA-ES and UC-DiffOSI outperform all baselines and are close to UC-Oracle.

**Evaluating Higher Number Dimensions.** To further compare UC-CMA-ES and UC-DiffOSI, we evaluate on the same wiping task where $\boldsymbol{\mu}$ is higher dimensional. We increase the number of DoFs with unknown stiffness and damping coefficients. The convergence curves can be found in supplementary Figure 6. In both the 2D and 3D optimization cases, UC-DiffOSI converges more quickly than UC-CMA-ES.

## 4.5 ROBUSTNESS TO VARYING DYNAMICS

In this example, we evaluate whether our proposed algorithm can handle varying dynamics throughout an episode. During each episode, we change $\bar{\boldsymbol{\mu}}$ at $t = 250$ (halfway through the episode) to a randomly sampled $\boldsymbol{\mu}' \sim [-0.6, 0.6]$. As shown in Table 4, our algorithm achieves notably better prediction accuracy than UC-MLP. However, the task performance is not as good as the oracle version. This is because in order to detect abrupt changes in the system dynamics, we need to collect sufficient data with the new system, which leads to a delay in identifying the correct parameters. For

Table 4: Results on the Cartpole task with changing $\mu$. We report errors on the parameter estimation of $\mu$ (lower is better), as well as the overall task performance (higher is better). Mean and standard deviation are reported. Results are averaged over 3 episodes.

| Approach | Mean Abs. Error ($\mu$) | Mean Abs. Rel. Error ($\mu$) | Task Performance |
|---|---|---|---|
| UC-MLP | $0.13 \pm 0.06$ | $0.92 \pm 0.84$ | $192.33 \pm 160.02$ |
| UC-DiffOSI | $0.02 \pm 0.01$ | $0.27 \pm 0.18$ | $222.33 \pm 89.09$ |
| UC-Oracle | $0.00 \pm 0.00$ | $0.00 \pm 0.00$ | $435.00 \pm 91.92$ |

Table 5: Results on sim-to-sim transfer (Nimble to Bullet) on the Cartpole task. We report errors on the parameter estimation of $\mu$ (lower is better), as well as the overall task performance (higher is better). Mean and standard deviation are reported. Results are averaged over 20 models and 100K timesteps.

| Approach | Mean Abs. Error ($\mu$) | Mean Abs. Rel. Error ($\mu$) | Task Performance |
|---|---|---|---|
| DR | N/A | N/A | $297.26 \pm 197.85$ |
| UC-DiffOSI | $0.02 \pm 0.04$ | $0.03 \pm 0.06$ | $434.10 \pm 110.88$ |
| UC-Oracle | $0.00 \pm 0.00$ | $0.00 \pm 0.00$ | $451.00 \pm 73.81$ |

abrupt and large changes in $\mu$ such as the one we used, the task performance can be sensitive to the accuracy of the prediction.

### 4.6 ROBUSTNESS TO NOVEL DYNAMICS MODELING

To evaluate the ability of our approach to generalize to environment variations beyond the training range, we apply our cartpole controller to the same task implemented in PyBullet physics engine. Due to differences in how the two physics engines solves the equation of motion and performs integration, a policy trained in UC-DiffOSI (Nimble) does not transfer directly to PyBullet. As shown in Figure 2b, our method still outperforms baselines. The gap is even larger between our method and DR, which suggests that in cases where the dynamics gap is larger, UC-DiffOSI can offer larger improvements.

## 5 CONCLUSION

UC-DiffOSI is a learning-based approach for training control policies that can operate in changing and unknown environments. Our method combines domain adaptation with a differentiable physics simulator by first training a Universal Controller that is conditioned on the environment parameters and then using a differentiable physics engine, NimblePhysics, to identify the environment parameters from a recent history of robot sensory inputs. By using a differentiable physics engine, we achieve efficient and generalizable system identification compared to prior methods based on learned models or traditional system identification. We evaluate our method on two robotic control problems: cartpole balancing and table wiping with a robot arm. Our method achieves superior performance than the baseline methods and is able to handle changing or un-modeled dynamics.

There are promising directions that further extend our work. For example, our algorithm currently assumes that the Universal Controller can optimally handle different environments given an accurate estimation of the environment parameters. However, in real applications, the robot might encounter situations that are beyond the capability of a pre-trained UC. Determining how to efficiently fine-tune a UC with limited data is thus important future work, where differentiable physics engines could play a pivotal role. In addition, this work focuses on rigid body environments, while real-world environments are filled with objects that can deform. An interesting direction could be to extend our approach to handle deformable objects while retaining high efficiency in the online system identification. Finally, we plan to apply our approach to real robot hardware, requiring bridging the sim-to-real gap and running the end-to-end control pipeline in real time.

## REPRODUCIBILITY STATEMENT

To maximize reproducibility, we describe our methodology in detail in Section 3 and our experimental setup in Section 4. The code, based upon on the open-source physics simulator (Nimble-Physics (Werling et al., 2021)), will be released upon publication to facilitate future research.

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

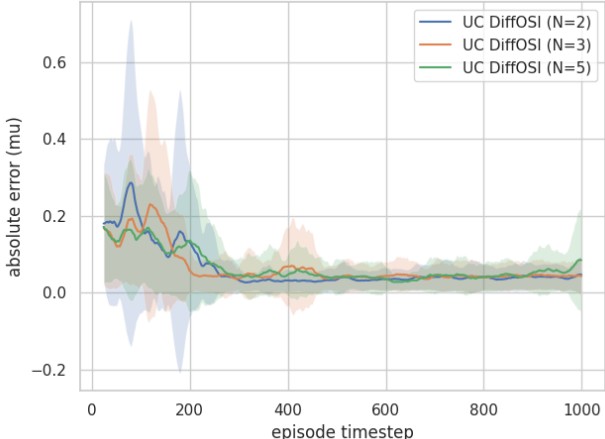

Figure 4: Analyzing the effect of the number of samples (window size from history) used by UC-DiffOSI vs. the absolute error $\mu$ prediction for the half cheetah task. $N$ denotes the context window size, and the number of samples required is $N - 1$.

## A  LATENCY

We additionally analyze the effect of the number of sampled used by DiffOSI on the prediction accuracy of $\hat{\mu}$. Results are shown in Figure 4. We compare different context window sizes (2, 3, and 5) and plot the absolute $\mu$ error as a function of the episode timestep for the half cheetah task. We find that a single sample (corresponding to a single step, with context window size of 2 containing states before and after the step) is sufficient to obtain an accurate prediction of $\mu$; more samples does not help. A single optimization step of DiffOSI takes 2-3 milliseconds.

## B  WIPING

In this section we show additional diagrams and results on the wiping task. Figure 5a and Figure 5b depict the wiping task as well as an overview of the system we implemented. Figure 6 shows a comparison between CMA-ES and DiffOSI in terms of absolute error $\mu$ predictions.

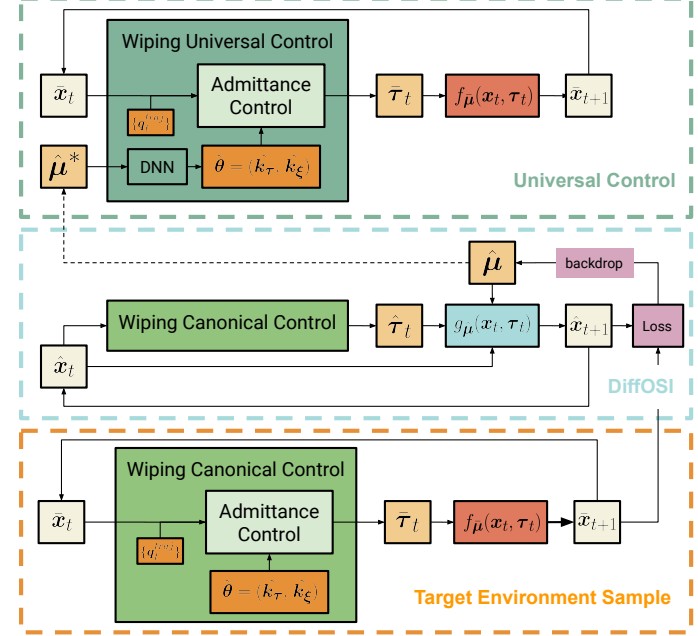

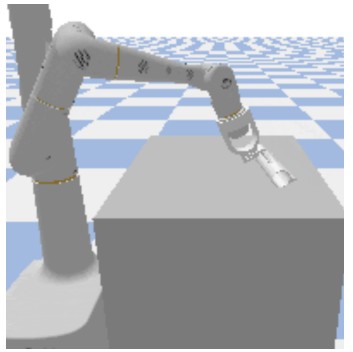

(a) Visualization of the wiping task performed in NimblePhysics simulation.

(b) Overview of applying our method, UC-DiffOSI, on the wiping task. From bottom to top, we first collect samples $\bar{X}$ in the target environment with a canonical wiping controller. Then, DiffOSI optimizes for the environment parameters $\hat{\mu}$ with the same controller. Finally, a pre-trained DNN-parameterized UC takes the identified $\hat{\mu}^*$ and generates optimal wiping control.

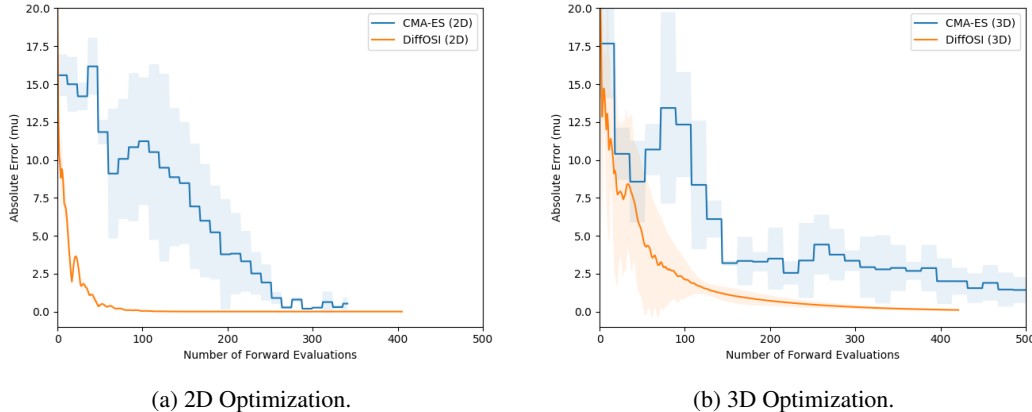

(a) 2D Optimization.

(b) 3D Optimization.

Figure 6: Comparison of CMA-ES and DiffOSI on the parameter estimation task. Results are averaged over 3 runs and all dimensions.

