# OpenReview forum: "Universal Controllers with Differentiable Physics for Online System Identification"
_ICLR.cc/2022/Conference — ICLR 2022 Submitted_

### Official Review · Reviewer_8qJx · 2021-10-27

**Correctness:** 2
**Technical Novelty And Significance:** 2
**Empirical Novelty And Significance:** 2
**Recommendation:** 5
**Confidence:** 4

**Main Review:**

The paper idea is overall interesting and fits very well with the current developments of differential simulation systems. The main strengths are:
1. It empirically shows that domain randomization is insufficient to solve complex control problems with varying dynamics. This is a surprising finding, which could help future research in robotics.
2. It shows that learning a universal control with a simple MLP architecture is possible if conditing it on robot parameters.
3. The paper is well written and easy to follow.

In my opinion, the major weakness of the current submission are:
1. The experimental results are not sufficient to back up the claims made in the introduction. Most of the experiments are done on the cart-pole system, which has been a benchmark of the control community for decades and is very simple to model and control. In the single experiment with a manipulator, the performance is slightly inferior to a baseline. Therefore, I would recommend performing experiments on a more diverse set of robots and tasks. The most interesting experiments are the ones in which the environment suddenly changes, where I think the proposed approach could have advantages over existing systems. Therefore, I would argue that sections 4.4 and 4.5 could be vastly extended.
2. There are no state-of-the-art model-based reinforcement learning baselines. They are very similar in their problem formulation (identifying a model and then using it for control via online or offline optimization). Therefore, I believe them to be valid baselines. Possible examples are PETS [R1] or Dreamer [R2]. As mentioned above, they would probably need many samples to adapt to changes, so this could be a selling point for the approach.

Other comments are:
* The approach used for table wiping is very similar to the one used for object cutting (in the real world!) proposed by [R3]. I think that this deserves at least mention and possibly quantitative comparison.
* It would be nice to have some demonstrator where differential simulators shine, e.g., for deformable objects.
* What would happen if the policy input were very high dimensional, for example, images? What would be the changes required to make the approach work?
* What about using a model-predictive controller (e.g., MPC) instead of a universal controller defined by an MLP? In theory, the MPC would require the same input of the MLP, i.e., the platform state and the robot parameters (i.e., the model). This could be very easily implemented, at least for the cartpole.

[R1] Deep Reinforcement Learning in a Handful of Trials using Probabilistic Dynamics Models, Chua et al.
[R2] Learning latent dynamics for planning from pixels, Hafner et al.
[R3] DiSECt: A Differentiable Simulation Engine for Autonomous Robotic Cutting, Heiden, et al.


**Summary Of The Paper:**

This paper proposes an algorithm to control robots with a universal controller conditioned on the robot parameters, that are identified online using differential simulation. The idea is simple yet interesting: giving the controller explicit information about the system could definitely help performance. Estimating those parameters with differential simulation definitely makes sense. The approach could be valuable for helping robots handle internal or external changes during operation. The approach is evaluated on a series of rigid body control tasks, where it is on pair or better than previous solutions.

**Summary Of The Review:**

The paper presents an interesting idea, but more experiments on different systems and tasks, and possibly a more diverse set of baselines, could significantly strengthen the submission.

---

> ### Author Response · Authors · 2021-11-23
> **Response to Reviewer 1**
>
> Thank you for your comments and suggestions on the paper!
>
> > I would recommend performing experiments on a more diverse set of robots and tasks.
>
> We agree with your suggestions, and have implemented a 7-link, 6-DoF legged robot in nimble, based on the 2D HalfCheetah task from OpenAI Gym. The experiment results can be found in Section 4.3 of the paper, with an additional Figure 3 and Table 2 containing results for this task. We show that UC generally outperforms pure domain randomization, and that UC-DiffOSI performs comparably to UC-Oracle while outperforming UC-MLP.
>
> > What about using a model-predictive controller (e.g., MPC) instead of a universal controller defined by an MLP? In theory, the MPC would require the same input of the MLP, i.e., the platform state and the robot parameters (i.e., the model). This could be very easily implemented, at least for the cartpole.
>
> We have added a comparison with MPC in the half cheetah experiment in Section 4.3. Because MPC is slower than UC, MPC is not a suitable method for a setting like ours where realtime performance is important.
>
> > What would happen if the policy input were very high dimensional, for example, images? What would be the changes required to make the approach work?
>
> To adapt our method to images, we would additionally need a module to transform from image pixel space to state space, since DiffOSI works in state space.

---

### Official Review · Reviewer_3zvY · 2021-11-02

**Correctness:** 4
**Technical Novelty And Significance:** 2
**Empirical Novelty And Significance:** 2
**Recommendation:** 5
**Confidence:** 4

**Main Review:**

Strong:

- The paper applies the recently developed differentiable physics and succeeds in applying it in a wiping task that requires contact-rich environment interaction. This will encourage the development of differentiable physics in the future.
- Combining differentiable physics with UCOSI is a natural idea and performs well in practice.

Weak:

- Applying differentiable physics for system identification has been seen in the previous work. Combining it with UCOSI method is rather straightforward. The paper's novelty is limited.
- Changing the environment from NimblePhysics to PyBullet doesn't mean good Sim2Real performance. The current paper only includes a very limited set of environments (cartpole and wiping), which is not enough to prove its effectiveness in practice.
- The number of parameters to identify in both environments is small. In this case, using finite-difference to compute gradients would be a not bad replacement for a differentiable physics engine. It would be better if the authors can consider environments with more dofs. For example, they can consider identifying the Inertia matrices of the robot arms.

**Summary Of The Paper:**

In this paper, the authors propose UC-DIFFOSI, which combines a differentiable physics simulator for system identification, and a universal controller which takes the identified parameters and passes them into the neural network to output actions. The method achieves better performance than domain randomization baselines and other system-identification methods.

**Summary Of The Review:**

The paper proposes a nice framework that combines differentiable physics with UCOSI. However, employing differentiable physics for system identification has only limited novelty. The experiments are a little bit weak with only 2 simulation tasks and no real-world environments. I think the paper's contribution doesn't pass the bar for acceptance.

---

> ### Author Response · Authors · 2021-11-23
> **Response to Reviewer 2**
>
> Thank you for your comments and suggestions on the paper!
>
> > The current paper only includes a very limited set of environments (cartpole and wiping), which is not enough to prove its effectiveness in practice.
>
> We agree with your suggestions, and have implemented a 7-link, 6-DoF legged robot in nimble, based on the 2D HalfCheetah task from OpenAI Gym. The experiment results can be found in Section 4.3 of the paper, with an additional Figure 3 and Table 2 containing results for this task. We show that UC generally outperforms pure domain randomization, and that UC-DiffOSI performs comparably to UC-Oracle while outperforming UC-MLP.

---

> > ### Comment · Reviewer_3zvY · 2021-11-30
> > **Response to Authors**
> >
> > Thanks for your response. However, adding a single domain does not make a substantial difference. Without addressing my remaining concerns, I will not change my score.

---

### Official Review · Reviewer_hY3g · 2021-11-02

**Correctness:** 3
**Technical Novelty And Significance:** 3
**Empirical Novelty And Significance:** 3
**Recommendation:** 6
**Confidence:** 4

**Main Review:**

The main contribution of the paper is the usage of a differentiable physics engine for online system identification, which supplements a universal controller(UC) trained offline.

The main concern is in its current form, the paper is not clear about the trade-off of the proposed approach. The results are not surprising: 1) Doing system ID is better than domain randomization and 2) Using a differentiable physics engine is better than using MLP prediction / blackbox optimization (e.g. CMA-ES) for system identification. Since either using UC nor using a differentiable physics engine for system ID is novel, while the combination of the two is what the paper is proposing, it can be clearer on the trade-off this combination offers. The paper could improve its quality if the following questions can be answered (1 is the major one while 2 and 3 are minor):

1. Latency: how fast is DiffOSI? Is there a trade-off between number of samples versus accuracy? How sensitive the task performance is with regard to latency (time spent on) in system ID?
2. Why are the number of runs/models different between table 1 (3 models) and figure 2 (20 runs).
3. Figure 5: 3 runs are not statistically significant. It's also a bit weird seeing CMA-ES achieves similar error around 200 evaluations for both 2D and 3D cases.

**Summary Of The Paper:**

The paper proposed to use differentiable physics engine for online system identification, which is then combined with controllers trained in a wide variety of tasks for robotic control problems in environment with unknown dynamics.

**Summary Of The Review:**

The idea of combining  a differentiable physics engine for online system identification with a universal controller(UC) is interesting but the paper could improve its quality by adding more details on how the two modules work for continuous control tasks (e.g. regarding latency).

---

> ### Author Response · Authors · 2021-11-23
> **Response to Reviewer 3**
>
> Thank you for your comments and suggestions on the paper!
>
> > Latency: how fast is DiffOSI? Is there a trade-off between number of samples versus accuracy? How sensitive the task performance is with regard to latency (time spent on) in system ID?
>
> Thanks for your questions! These are indeed important details. We added discussion on latency in Section A of the Supplementary Material. We also added a plot that shows the tradeoff between number of samples vs. accuracy, over the course of episode timesteps. To summarize, a single sample (context window size=2) is generally sufficient to obtain good accuracy. A single optimization step of DiffOSI takes 2-3 milliseconds.

---

### Official Review · Reviewer_S8au · 2021-11-05

**Correctness:** 4
**Technical Novelty And Significance:** 2
**Empirical Novelty And Significance:** 2
**Recommendation:** 5
**Confidence:** 2

**Main Review:**

Strengths:

This paper takes advantage of advances in differentiable physics (e.g., Werling et al. (2021)) and combines this with the idea of Universal Controllers (Yu et al., (2017)) to build controllers that are more robust. The proposed approach outperforms baselines in two benchmarks.

Weaknesses:

This paper seems to only differ from Yu et al., 2017 in that it replaces the MLP for OSI with a differentiable physics simulator. Given that the paper easily combines already published ideas, I would expect rich experimental results. They use two tasks of cartpole balancing and tabletop wiping. Other works have more complex domains such as 4-link, 6-DoF 2D hopper and the 18-DoF Unitree Laikago quadruped in Jiang et al., (2021)  and Fetch robot in Peng et al., (2017).

Minor:

Some figures are not referenced in the text (e.g., Figure 3 and 4, in Figure 2 only 2b is referenced).

**Summary Of The Paper:**

The paper tackles the problem of learning robot controllers that can handle changing or unknown environments. It proposes to use differentiable physics for online system identification and reinforcement learning for offline policy training. The differentiable physics module estimates simulation parameters from robot history and feeds this to the controller that is parameterized by these simulation parameters. They use domain randomization to ensure the universal controller conditioned on simulation parameters is robust to changing environments (simulation parameters). At test time, the differential physics simulation is used to estimate the simulation parameters to bias the controller to output controls for the 'correct' simulation parameters.

The proposed approached is evaluated against several benchmarks on the cartpole and a tabletop wiping tasks and has been shown to outperform the baselines, which include domain randomization and some domain adaptation approaches.

**Summary Of The Review:**

Given minimal modifications from existing work, this work is missing comprehensive experiments. Experiments on real robots as mentioned as part of future work would strengthen the paper significantly.

---

> ### Author Response · Authors · 2021-11-23
> **Response to Reviewer 4**
>
> Thank you for your comments and suggestions on the paper!
>
> > I would expect rich experimental results. They use two tasks of cartpole balancing and tabletop wiping. Other works have more complex domains such as 4-link, 6-DoF 2D hopper and the 18-DoF Unitree Laikago quadruped in Jiang et al., (2021) and Fetch robot in Peng et al., (2017).
>
> We agree with your suggestions, and have implemented a 7-link, 6-DoF legged robot in nimble, based on the 2D HalfCheetah task from OpenAI Gym. The experiment results can be found in Section 4.3 of the paper, with an additional Figure 3 and Table 2 containing results for this task. We show that UC generally outperforms pure domain randomization, and that UC-DiffOSI performs comparably to UC-Oracle while outperforming UC-MLP.
>
> > Figures should be referenced in the text (e.g., Figure 3 and 4, in Figure 2 only 2b is referenced).
>
> Thanks. We have added references and discussion to all figures.

---

### Decision · Program_Chairs · 2022-01-20

**Decision:**

Reject

**Comment:**

The paper addresses the learning of robot controllers for changing or unknown environments. It makes use of differentiable physics for online system identification and of reinforcement learning for offline policy training. A universal controller is trained on a distribution of simulation parameters in order to ensure its robustness. Differentiable physics is used to estimate the simulation parameters from the recent observation history. These parameters are fed to the controller so as to modulate the policy. This approach is evaluated on three benchmarks (2 + 1 added during the rebuttal).

The main originality of the paper is the use of differentiable physics for the identification of the parameters in the context of varying environments. The topic is of interest and in line with the recent developments for robotics. However, the novelty is limited, and all evaluators were concerned about the limited experimental contribution. The authors have added a new experiment during the rebuttal but this was not considered sufficient to change the evaluation. Overall this is considered as a promising contribution, but the experimental setting should be largely improved with additional problems and comparisons with SOTA methods from the recent RL literature.